# Type 1 diabetes and parasite infection: An exploratory study in NOD mice

Emilie Giraud[1], Laurence Fiette[2], Evie Melanitou[3]*

**1** Chemogenomic and Biological Screening Core Facility, C2RT, CNRS UMR 3523, Institut Pasteur, Université Paris Cité, Paris, France, **2** Human Histopathology, and Animal Models Laboratory, Institut Pasteur, Université Paris Cité, Paris, France, **3** Department of Parasites & Insect-Vectors, Institut Pasteur, Université Paris Cité, Paris, France

* evie.melanitou@pasteur.fr

## Abstract

Microorganisms have long been suspected to influence the outcome of immune-related syndromes, particularly autoimmune diseases. Type 1 diabetes (T1D) results from the autoimmune destruction of the insulin-producing beta cells of pancreatic islets, causing high glycemia levels. Genetics is part of its aetiology, but environmental factors, particularly infectious microorganisms, also play a role. Bacteria, viruses, and parasites influence the outcome of T1D in mice and humans. We used nonobese diabetic (NOD) mice, which spontaneously develop T1D, to investigate the influence of a parasitic infection, leishmaniasis. *Leishmania amazonensis* is an intracellular eukaryotic parasite that replicates predominantly in macrophages and is responsible for cutaneous leishmaniasis. The implication of Th1 immune responses in T1D and leishmaniasis led us to study this parasite in the NOD mouse model. We previously constructed osteopontin knockout mice with a NOD genetic background and demonstrated that this protein plays a role in the T1D phenotype. In addition, osteopontin (OPN) has been found to play a role in the immune response to various infectious microorganisms and to be implicated in other autoimmune conditions, such as multiple sclerosis in humans and experimental autoimmune encephalomyelitis (EAE) in mice. We present herein data demonstrating the role of OPN in the response to *Leishmania* in NOD mice and the influence of this parasitic infection on T1D. This exploratory study aimed to investigate the environmental infectious component of the autoimmune response, including Th1 immunity, which is common to both T1D and leishmaniasis.

## Introduction

Genetic predisposition and environmental triggers are etiological factors for autoimmune conditions. The incidence of autoimmune disease is increasing in industrialized countries, while in contrast, clean environments and vaccination policies can be used to control infections. The "hygiene" or "old friends" hypothesis proposes that vaccination and the eradication of infectious microorganisms in these countries may contribute to an excessive autoimmune response against self-tissues and molecules by interfering with the immune system's normal function

**Data Availability Statement:** All relevant data are within the paper and its Supporting Information files.

**Funding:** This study benefitted from financial support provided by the Institut Pasteur in the form

of functional overheads (to EM for experiments and the animal care) as well as salaries inclusive for all authors. There was no additional specific funding received for this study.

**Competing interests:** The authors declare that they have no competing interests.

[1–5]. In support of this hypothesis, epidemiological data show a North–South gradient of autoimmune disease frequency and opposite geographical distribution to infectious conditions [6, 7].

The evolutionary adaptation between hosts and microorganisms maintains a homeostatic immune equilibrium requiring the presence of both. The Red Queen hypothesis suggests that genetic mutations promote organisms' constant adaptation for survival against ever-evolving opposing species. This hypothesis exemplifies antagonistic interactions between parasites and the host, allowing coevolution dynamics to occur [8, 9]. Constant adaptation of the host genome to environmental triggers, including infectious microorganisms, may be the origin of genomic and/or epigenomic changes required for host survival. Immune function genes are at the forefront of such adaptation, and in the absence of infections, host immune system dynamics may functionally affect immune homeostasis, leading to autoimmune, non-self-recognition. Infections play a significant role in the environmental component of the etiology or even protection against autoimmune diseases [10].

Type 1 diabetes (T1D) is an autoimmune condition in which insulin-secreting beta cells of the pancreas are affected, resulting in hyperglycemia. It represents 7–12% of the total cases of diabetes in the world, estimated at 425 million people worldwide [11]. The frequency of T1D is greater in urban than in rural environments [12]. As urbanization expands, its incidence increases [11]. Its genetic origin is well established, and over 20 genes have been identified as part of the inherited component conferring up to 50–80% of T1D etiology [13, 14]. However, genetics alone cannot account for the rapid expansion of its incidence in the last 100 years [15–18]. Environmental factors, including infections by various microorganisms, bacteria, viruses and parasites, play a role in modulating autoimmunity. Viral, bacterial, and parasitic infections reportedly elicit or protect against disease pathogenesis [18]. Lymphocytic chorio-meningitis virus (LCMV) has been reported to play a protective role against T1D in nonobese diabetic (NOD) mice [19, 20], while epidemiological data suggest that enteroviruses such as Coxsackie virus B (CVB) are involved in the initiation or acceleration of this disease in humans [20–22]. It was shown that parasitic infections caused by helminths prevent diabetes and ameliorate insulin secretion [23–25]. Similarly, schistosomiasis protects against diabetes [25], as does the TB (tuberculosis) vaccine Bacillus Calmette-Guerin (BCG), which induces a host response preventing T1D [26, 27]. The increase in diabetes mellitus in subtropical countries is accompanied by the attenuation of immune defenses and an increase in susceptibility to infections due at least in part to the high levels of uncontrolled glycemia most of the time [28].

Leishmaniasis, a neglected disease group with a high prevalence worldwide, is caused by an intracellular protozoan parasite, *Leishmania spp.* [29]. Diabetes can worsen the outcome of cutaneous lesions caused by *Leishmania (L.) major* or *L. infantum* parasites [30]. Similarly, diabetes enhances ulcerated lesions caused by *L. braziliensis* by affecting proinflammatory cytokine levels and impairing the response to therapy [31]. *L. amazonensis* (*L. am.*) causes the cutaneous form of the human disease, and it is transmitted by the bite of infected phlebotomine sandflies during blood feeding [32–34].

The role of osteopontin (OPN) in the host response to *L. amazonensis* has been evaluated in our previous studies [35]. OPN, encoded by the *spp1* gene (secreted phosphoprotein 1) originally identified in activated T cells (also named *eta-1* for the early T-cell activation-1 gene), is a key cytokine implicated in an efficient type 1 immune response (Th1) and differentially regulates *il-12* and *il-10* expression in macrophages (MFs) [36]. Mice in which the *spp1* gene is knocked out exhibit severely impaired Th1 immunity to viral and bacterial infections [36]. OPN is also associated with autoimmune diseases, including T1D, in humans and mice [37–39]. Moreover, OPN is an early marker for islet autoimmunity in human T1D patients (our unpublished data and [40]). We have previously reported that OPN is involved in the host

response to *L. am.* parasites in C57BL/6 mice [35]. These parasites induce *opn* gene expression in bone marrow-derived macrophages (BMFs) *in vitro* and *in vivo* at the sites of parasite inoculation and inhibit proinflammatory transcripts [35].

The role of OPN in the immune system in both infectious and autoimmune diseases led us to investigate its role in the infectious environmental component of autoimmune disease etiology, as postulated by the "hygiene" hypothesis. We generated mutant mice lacking the osteopontin gene in a NOD genetic background [39] and demonstrated that in the absence of this gene, T1D is accelerated, suggesting a protective effect of OPN on the disease [39].

Our study of wild-type and *opn* knockout NOD mice revealed that infection with *L. am.* has differential effects on the T1D phenotype and parasitic load, depending on *opn* expression. We also observed variations in clinical phenotype, parasite content and proinflammatory markers in the absence of OPN. After infection with the parasites, a Th1 to Th2 shift was observed in the absence of OPN.

## Materials and methods

### Animals and ethics statement

NOD/*Shi*LtJ mice were used at 6–8 weeks of age (referred to as wild-type or NOD$^{+/+}$) and were purchased from Jackson Laboratories (Bar Harbor, ME, United States). *Opn* mutant congenic mice (NOD.B6.Cg.*opn*$^{-/-}$ for simplicity, referred to as NOD KO or NOD.*opn*$^{-/-}$) were bred in our facilities [41]. All animals were housed in the animal facility of the Institut Pasteur, and all experimental procedures were approved by the Institutional Committees on Animal Welfare at the Institut Pasteur (n˚: 2013–0014) and carried out under strict accordance with European guidelines (Directive 2010/63/EU). Experimental procedures were carried out under strict respect of the 3R (Replacement, Reduction, Refinement) and in strict accordance with European guidelines (Directive 2010/63/EU). Disease prevention and consideration of discomfort levels were carefully assessed throughout the experimental procedures. For euthanasia we used carbon dioxide ($CO_2$) inhalation with increasing concentrations of $CO_2$, mice were housed in groups before euthanasia and kept in their home cage. Animal experimentation was conducted by EM and EG, both of whom are authorized by the Paris Department of Veterinary Services, DDSV.

### Preparation and inoculation of *L. amazonensis* parasites

The parasites were prepared for inoculation following previously described methods [42–44]. Specifically, $10^6$ *L. amazonensis* strain LV79 amastigotes (WHO reference: MPRO/BR/72/M1841) were subcutaneously inoculated into the hind footpads of Swiss nude mice. After 2 months, the lesions containing the parasites were excised and purified according to established procedures [42, 45]. BMF cultures were infected with LV79 amastigotes at a MOI (multiplicity of infection) of 4:1. LV79 metacyclic promastigotes carrying the firefly luciferase gene in the 18S rRNA locus of *L. amazonensis* LV79 strain nuclear DNA [42] were prepared from amastigotes and cultured at 26˚C, as previously described [46]. Infective-stage metacyclic promastigotes were obtained from 6-day-old stationary phase cultures by Ficoll gradient, and $10^4$ metacyclic promastigotes were injected into the ear dermis of NOD$^{+/+}$ and NOD.*opn*$^{-/-}$ mice. The mice were anaesthetized by intraperitoneal administration of ketamine (120 mg/kg Imalgène 1000, Merial, France) and xylazine (4 mg/kg; Rompun 2%, Bayer, Leverkusen, Germany). Clinical phenotypes were assessed by evaluating the range of the lesion size at the site of inoculation and compared to that of the non-inoculated contra-lateral ear, as previously described [42]. Ear thickness was measured using a Vernier calliper (Thomas Scientific, Swedesboro, NJ).

## Bone marrow-derived macrophage generation and infections

To obtain macrophages, bone marrow cells were extracted from the tibia and femurs of 6- to 8-week-old NOD[+/+] and NOD.*opn*[-/-] mice and differentiated into macrophages (BMFs) using established methods [45]. Briefly, the bone marrow cells were suspended in PBS-Dulbecco enriched with $Ca^{++}$ and $Mg^{++}$, collected by centrifugation and cultured in the presence of rm-CSF-1 (ImmunoTools) at a density of $7.5 \times 10^6$ cells/100 mm Falcon dish and maintained at 37˚C (94% air, 7.5% $CO_2$). *L. amazonensis* amastigotes (strain LV79) were freshly isolated from footpad lesions of Swiss nude mice as previously described [47] at a MOI of 4:1 (parasites: MF) on day 7. The cells were incubated at 34˚C for 24 or 48 hours before being lysed for total RNA preparation or placed on glass microslides for fluorescent immunostaining with antibodies against LAMP-1, OPN and *Leishmania*.

## Flow cytometric analysis of macrophage-restricted marker expression

Flow cytometry was used to examine differentiated macrophages from NOD wild-type and NOD *opn* knockout mice. Monocyte/macrophage lineage surface molecules, including CD11b, CD115, CD11c, MHC-II and the F4/80 antigen, were evaluated using the corresponding antibodies [48, 49].

FACS analysis revealed that 98% of the cells exhibited similar characteristics, with CD11b[High], F4/80[High], and CD115[High,] while surface-specific markers for the dendritic cell (DC) lineage were negative for CD11c[-] and high for MHC-II[High] (S1 Fig). The data were analysed using Kaluza software on a Gallios flow cytometer from Beckman Coulter [45].

## *In vivo* bioluminescence imaging of luciferase-expressing *L. amazonensis*

Animals were monitored for clinical phenotypes and parasite proliferation at various time points ranging from day 16 to day 100. Parasite loads were assessed using bioluminescence imaging as previously described [42]. Briefly, luciferin was injected into animals (150 mg/kg i. p., D-Luciferin potassium salt, SYNCHEM OHG, Germany), and photons emitted from the entire ear pinna (ROI) were acquired by a camera. The same ROI was examined for all mice at all time points. Total photon emission was expressed in photons/sec/ROI. The median bioluminescence values and SDs were calculated for each experimental group. At specific time points, 3 representative mice from each group were sacrificed for further analysis [50]. Contralateral tissues from the noninjected and control mouse groups were analysed in parallel.

## Immunofluorescence labelling of BMFs

BMFs were cultured on glass slides (CML France) and either infected or not infected with *L. amazonensis* amastigotes (MOI 4:1) at 34˚C (94% air, 7.5% $CO_2$). After 24–48 hours, the cells were washed with PBS (Dulbecco), fixed with 4% paraformaldehyde (PFA) for 1 h at room temperature and permeabilized with saponin (25 mg/ml). Then, the cells were labelled with the following primary antibodies: 10 μg/ml amastigote-specific mAb 2A3-26-biot, 7 μg/ml anti-OPN (goat IgG) (R&D Systems, France) or LAMP-1/CD107a monoclonal antibody specific for lysosomal-associated membrane protein 1 (LAMP-1) of the parasitophorous vacuole (rat IgG2a FITC, Invitrogen, CA). Revelation was performed with 1.5 μg/ml streptavidin conjugated to Texas Red (Molecular Probes, Cergy Pontoise, France) for the *Leishmania* parasites, with donkey anti-goat FITC (fluorescein isothiocyanate fluorochrome, sc2024, Santa Cruz Biotechnology) for osteopontin and donkey anti-rat-FITC (LS-C351178, CliniSciences, FR) for LAMP-1. After incubation with the primary antibodies (30 min) and three washes with PBS/saponin, a second incubation was carried out for 30 min with the secondary antibodies. Glass

slides were then mounted with Hoechst 33342-containing Mowiol 4.88 (Calbiochem), allowing visualization of the DNA of both host cell and amastigote nuclei. Epifluorescence microscopy was used to detect the signals, and the mean protein densities were analysed with a Zeiss AxioVision Rel. 4.8.2 Image acquisition software (Carl Zeiss International) At least 10 different cells were analysed in at least three different fields. The values are expressed in density units/msec of $20\mu m^2$ *opn*-stained areas/cell. Statistical analyses were performed with the Mann–Whitney test.

## Histopathology

Samples of the ear pinna, draining lymph nodes and foot pads were fixed in 4% formalin, embedded in paraffin, and stained with hematoxylin and eosin. Microscopic changes were scored semi-quantitatively by using a five-scale scoring system (1: minimal, 2: mild; 3: moderate, 4: marked, 5; severe) for parameters such as ulceration, acanthosis, necrosis, oedema, and the presence of parasites. The median values were compared between the inoculated and control tissues. Immunostaining with anti-OPN (AF808, R&D Systems, Minneapolis, MN, USA) and anti-F4/80 macrophage-specific antibodies (MAB 5580, R&D Systems) was performed to evaluate the infiltration of neutrophils and macrophages.

## RNA Extraction and real-time quantitative PCR

Tissues from representative mice at different time points and $5x10^6$ BMFs were lysed for RNA preparation using the RNeasy Plus Mini Kit (Qiagen, SAS, France) following the manufacturer's instructions, as previously described [50]. RNA quality and quantity were evaluated by measuring the optical density using a Nanodrop ND-1000 micro-spectrophotometer (Thermo Fisher Scientific) [51].

Real-time quantitative PCR (qRT-PCR) was performed following a previously described protocol [50]. The RNAs were reverse transcribed using random hexamers (Roche Diagnostics) and MMLV-RT reverse transcriptase (Moloney Murine Leukemia Virus, Invitrogen Life Technologies). The relative quantification of the genes of interest was carried out in a 10 μl reaction volume in white UltraAmp 384-well PCR plates (Sorenson, Bioscience, Salt Lake City, UT, USA) using QuantiTect SYBR Green (Qiagen) and a LightCycler® 480 system (Roche Diagnostics, Meylan, France). The primers were used at a final concentration of 0.5 μM (Guaranteed Oligos™, Sigma–Aldrich), and the PCR program consisted of 40 cycles of denaturation at 95˚C for 10 sec, annealing at 54˚C for 25 sec and extension at 72˚C for 30 sec.

The SYBR Green fluorescence emission was measured at the end of the elongation step, and the crossing point (Cp) values were determined using the second derivative maximum method of LightCycler® 480 Basic Software. The qbase program (Biogazelle qbase) for qPCR data management and analysis was used to analyse the raw Cp values [52]. Eleven candidate control genes were tested as previously described with the geNorm [53] and NormFinder [35, 54] programs. *Hprt* and *ywhaz* were selected as the reference genes for normalizing the gene expression levels (for primers, see S1 Table). The relative expression levels of the genes of interest were calculated, and *Leishmania* parasites were quantified with the gene target primers (*ssrRNA*) F-CCATGTCGGATTTGGT and R- CGAAACGGTAGCCTAGAG [55].

## Statistical analysis

The number of parasites and their relative quantification in the BMF cells were determined using the "Mean parasite intensity", which is the mean number of parasites per infected host cell. The non-infected cells were not taken into consideration. "Prevalence", on the other hand, is the percentage of infected cells and provides information on the relative sizes of the

cells in the study (infected and uninfected). "Crowding" is a measure of parasite density and is defined as the sum of crowding values (parasites living in a cell) divided by the number of parasites [56]. While "mean intensity" is the sum of parasites per infected host, "mean crowding" represents the average number of parasites that can live in a host cell [57]. To compare the parasite load, mean intensity and crowding between macrophages from NOD$^{+/+}$ and NOD. $opn^{-/-}$ strains, quantitative parasitology (QP3.0) statistical software was used [58]. Mean intensity comparisons were performed by a bootstrap test [59], and the density-dependent character of parasites was described by crowding, taking p values at Cl (95% confidence limit) [59].

For the statistical analysis and graphs, GraphPad PRISM T.0 (GraphPad Software, San Diego, CA) was used. The Mann–Whitney test was used to compare relative transcript expression between macrophages from NOD$^{+/+}$ and NOD.$opn^{-/-}$ BMFs. The p values were calculated with a Mann–Whitney test to compare ear bioluminescence, ear width, and relative transcript expression obtained by qRT–PCR between the control group and the group infected with parasites (*: $p<0.05$; **: $p<0.001$; ***: $p<0.0001$).

## Results

### OPN–parasite interactions and T1D

Longitudinal analysis of the cumulative incidence of T1D in NOD wild-type (NOD$^{+/+}$) and *opn* knockout (NOD.$opn^{-/-}$) mice was carried out in the presence or absence of *L. am.* parasites (Fig 1A and 1B). Survival curves showed that in the absence of OPN (NOD.$opn^{-/-}$), T1D is

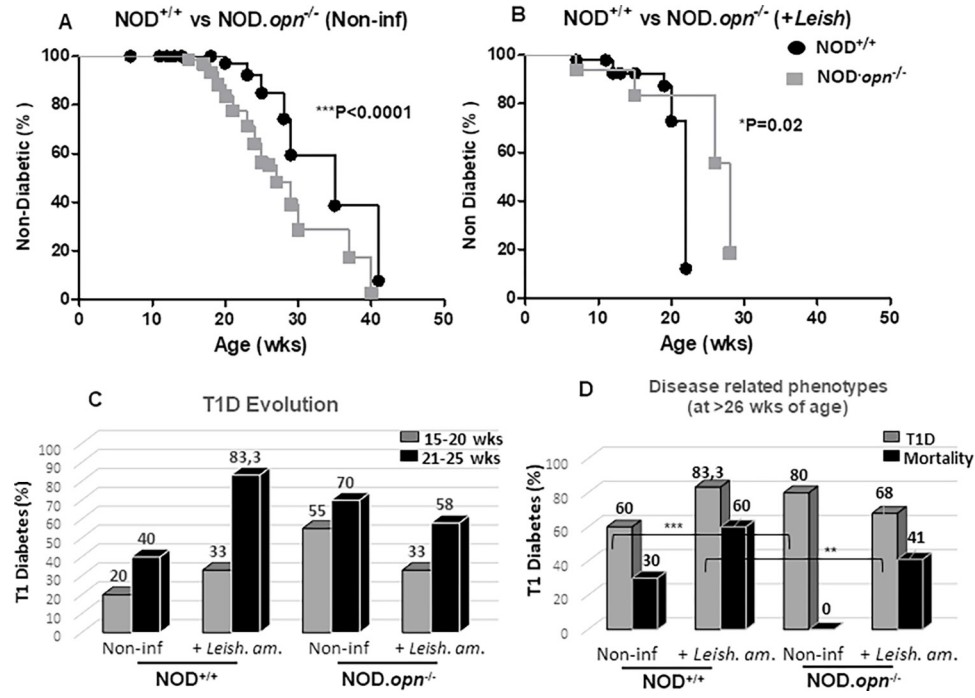

**Fig 1. OPN-parasite interactions in the T1D phenotype in NOD mice.** A & B. Longitudinal analysis of T1D (accumulative incidence, survival %) in (A) non-infected NOD (NOD$^{+/+}$ n°:10) and *opn* mutant (NOD.$opn^{-/-}$ n°:20) mice (P<0.0001). (B) T1D survival (%) in animals infected with *L. amazonensis* parasites at early preinflammatory stages (6–8 weeks). NOD.$opn^{-/-}$ n°: 12; NOD$^{+/+}$ INF n°: 6 (P = 0,0260). (C) T1D evolution (%). The cumulative incidence was calculated at two phenotypic windows: 15–20 weeks and 21–25 weeks of age. (D) Disease-related phenotypes, comparative analysis at 26 weeks of age. T1D in NOD$^{+/+}$ vs NOD.$opn^{-/-}$ non-infected (P = 0.0004) and animals infected with *Leishmania amazonensis* NOD$^{+/+}$ vs NOD.$opn^{-/-}$ (P = 0.016). (log-rank (Mantel-Cox) test) (see also S2 Table).

accelerated in comparison with the wild-type animals (Fig 1A, P<0.0001). In contrast, after infection with the parasites, T1D was significantly delayed in the absence of OPN (Fig 1B, P = 0.0260). T1D progression between the infected and noninfected animals was compared, taking into consideration the cumulative disease frequencies at two phenotypic windows according to age, at 15–20 and 21–25 weeks. Although the breeding conditions of the NOD colonies influence T1D, glycemia usually appears after 20 weeks of age, while most of the animals become diabetic at approximately 30 weeks [60, 61]. In the absence of OPN, animals develop accelerated diabetes between 15–20 weeks of age (55%), while in contrast, at this age, a low diabetes incidence was observed in wild-type animals (20%) (Fig 1C) [39]. In the presence of the *L.am.* parasites, while the early incidence of T1D (15–20 weeks) remained low, in the wild-type animals (33%), the cumulative incidence after 21 weeks increased (83.3%, Fig 1C). In the absence of OPN, *L. am.* infection inhibited T1D in both age groups (33% and 58%, respectively) (Fig 1B and 1C). Moreover, increased mortality at more than 26 weeks of age was observed in the infected animals for both wild-type (60%) and *opn*-knockout (41%), while in the absence of parasites, 0% mortality was detected in the opn-knockout animals (Fig 1D and S2(A) Table). These data indicate that in the absence of *opn*, an immune balance takes place favouring survival, despite high blood sugar levels, whereas in the presence of *Leishmania*, cumulative autoimmune- and infection-related morbidity occurs (Fig 1D and S2(B) Table). However, the waning role of OPN remains unclear. Indeed, while the presence of this protein in non-infected animals is protective against T1D (Fig 1C, 40% of NOD$^{+/+}$ versus 70% of NOD.*opn*$^{-/-}$), its absence does not favour an additional deleterious phenotype (morbidity 30% vs 0%, respectively) (Fig 1D). Although the evolution of body weight during aging in both WT and KO mice was similar after infection with the parasites (S2B Fig), lower body weights were systematically observed in the absence of OPN (S2A Fig). This finding indicates that OPN contributes to the general physiological status of the animals and is consistent with the protective role of OPN against T1D in NOD mice (Fig 1A), as reported previously [39]. However, this effect is abolished in the presence of parasites, as shown by a higher number of diabetic animals (83% in infected vs 40% in non-infected animals). In the absence of OPN, *L. am.* infection protects against T1D (Fig 1C and S2C Fig), probably by interfering with the balance of the Th1 responses elicited by parasitic infection. We hypothesized that in the absence of OPN, the Th1/Th2 paradigm of resistance/susceptibility of the host response may shift toward an aggressive for the parasites, Th2 immune phenotype.

### *In vivo* infection of NOD mice with *L. amazonensis* metacyclic promastigotes

**Clinical phenotype and parasitic load.** To address our hypothesis that the absence of OPN impacts the Th1/Th2 immune balance, which favours the persistence of parasites and contributes to lower glycemia, we performed additional *in vivo* experiments via a multiparametric approach to delineate distinct phases after the inoculation of *L. am.* We assessed the *in vivo* responses of the NOD mice to the parasites in the presence or absence of OPN. NOD wild-type and *opn* knockout mice were inoculated in the ear dermis with luciferase-expressing *L. am.* metacyclic promastigotes ($10^4$), the infectious form of *L. am.* Longitudinal analysis of the clinical phenotype (ear width) revealed variations between the mice, with a significant acceleration of lesion development in the absence of OPN starting on day 32 post-infection (*p. i.)* (Fig 2A).

Consistent with the clinical phenotype, the parasitic load was greater in the *opn* mutant mice, remaining at high levels after day 70 *p.i.*, while the parasite load of the WT animals significantly decreased (Fig 2B).

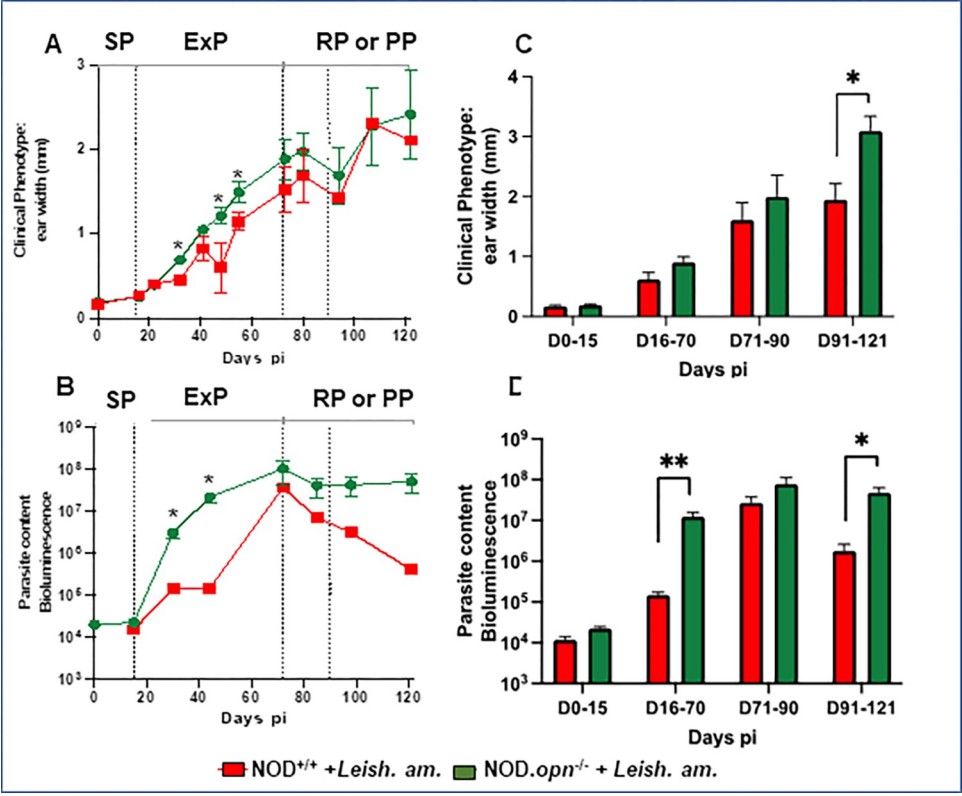

**Fig 2. *In vivo* clinical phenotype evaluation and parasitic load.** NOD/LtJ (NOD[+/+]) wild-type and *opn* mutant mice (NOD.*opn*[-/-]) were inoculated with $10^4$ *L. amazonensis* metacyclic promastigotes in the ear pinna. (A) Longitudinal examination of the clinical phenotype (ear width) at different time points post-inoculation. Medians and SDs are indicated. Ear width differences between *the opn* knockout and wild-type mice were observed at >30 days *p.i.* (P = 0.0017). The X-axis represents days *p.i.* The four phases of infection (SP, ExP, RP and PP) are described in the text. (B) Monitoring of fluctuations in the parasite load by bioluminescence. Signals are captured in the dermis of ear pinnae. The results obtained from wild-type mice (NOD[+/+]: n°: 15 mice per group) are represented in green, and those from *opn* mutant mice (NOD.*opn*[-/-], n°: 14 mice per group) are represented in red. (C) and (D) Time windows (days) after inoculation of $10^4$ metacyclic promastigotes of luciferase-expressing *L. amazonensis* in the ear dermis of wild-type and *opn* knockout NOD mice. Animals were grouped for each time window as indicated on the X-axis up to 120 days *p.i.* (WT: n = 15 mice, KO: n = 14 mice). (C) Comparison of clinical lesions at different time points post inoculation (ear widths) (Mann–Whitney test; *P < 0.017). (D) Comparison of the parasitic load determined by bioluminescence signal quantification at different time points post inoculation (the Medians and SDs are indicated). (Mann–Whitney test; D16–70 **P = 0.0046; D91–121 *P = 0.0382).

Overall, the clinical phenotype, followed for more than 100 days post-inoculation, progressed through four discrete phases (Fig 2C and 2D). The first phase (window D0-15 *p.i.*) is the silent phase (SP), during which neither parasites nor clinical features can be detected (Fig 2A and 2B). Then, the expansion phase (ExP) of *L. am.* was associated with an inflammatory aspect of the cutaneous lesion developing into erythematous oedema (window D16-70 *p.i.*). The parasitic load indeed increased over time in both wild-type and *opn* mutant mice, and this increase was significantly higher and appeared earlier in the absence of OPN (Fig 2B and 2D). Then, the parasitic load reached a peak (window D71-90 *p.i.*). The last phase is either a reduction phase (RP) for the WT mice, associated with a significant reduction in parasite load (Fig 2B and 2D), or a plateau phase (PP) for the KO mice, associated with the maintenance of parasite load at a high level (window D91-121 *p.i.*). Histological evaluation of the infected ear tissues revealed discrete differences in the microscopic lesions between the wild-type and *opn* mutant mice at 100 days *p.i.*, corresponding to the last phase (Fig 3A and 3B). The parasite

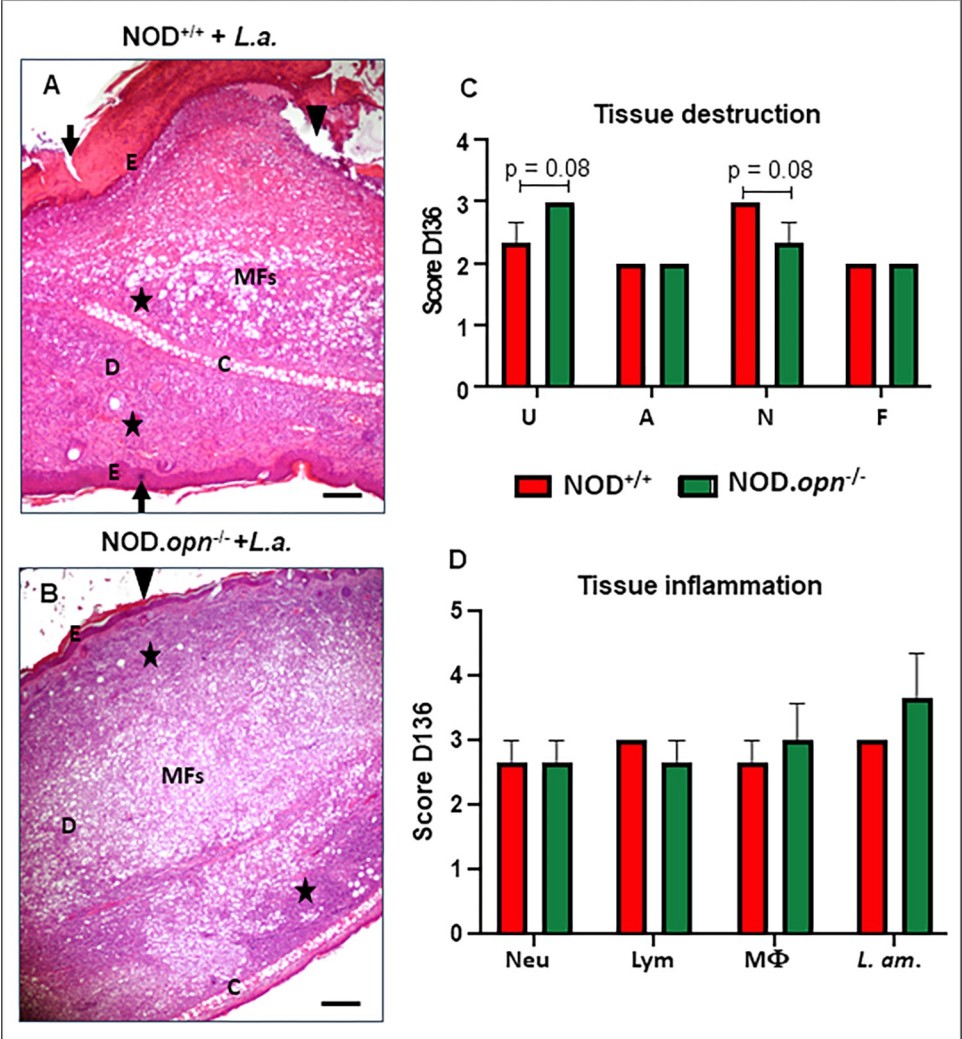

**Fig 3. Histopathology and evaluation of tissue inflammation markers at the site of parasite inoculation.** (A) Histological features of ear lesions in wild-type (NOD$^{+/+}$, n°: 5) and (B) *opn* mutant mice (NOD.*opn*$^{-/-}$, n° = 6) at 136 days *p.i.* (age 25 weeks). Microscopic lesions showed discrete differences between wild-type and knockout mice: parasites and MF content were higher in the absence of OPN. Triangles: ulceration, *stars: inflammation, infected macrophages, arrow: acanthosis: E: epidermis, D: dermis, C: cartilage. Hematoxylin and eosin (H&E), original magnification x4, scale bar: 100 μm. (C) and (D). Histological scores of NOD$^{+/+}$ (red bars) and NOD.*opn*$^{-/-}$ mice (green bars) post-inoculation with $10^4$ *L. amazonensis* metacyclic promastigotes. (C). Tissue destruction and (D). Tissue inflammation. (C) and (D) Abbreviations: U = ulceration, A = acanthosis, N = necrosis, F = fibrosis, Neu = neutrophils, Lym = lymphocytes, MF = macrophages, *L. am.* = *L. amazonensis*. Two-way ANOVA was used for statistical analysis.

content was greater in the absence of OPN, as was the number of macrophages (Fig 3D and S3C & S3D Fig). At this time point, despite the rather discrete differences in the impact of the *Leishmania* parasites in both wild-type and *opn* mutant mice, the histological examination of the infected sites confirmed a higher content of parasites (S3D Fig) as well as pronounced ulceration in the *opn* knockout mice (Fig 3C), while necrosis was more pronounced in the presence of *opn* (Fig 3C). Differences in the impact of parasite infection related to OPN were evident at the earlier phases after infection (Fig 2A and 2B).

**In vivo host response to the parasites and role of OPN: Comparative analysis between NOD and C57BL/6 mice.** On the C57BL/6 genetic background, we have previously shown

an increase in inflammatory lesions after *L. am*. infection in the ear pinna of the *opn* knockout in comparison with the wild-type mice, while in contrast, the parasite content remained similar [35].

As mentioned above, in the NOD mice, four phases were delineated: a silent phase (SP), an expansion phase (ExP), a pick phase (PP) and a reduction (RP) (in the WT mice) or a plateau phase (in the KO mice) (Fig 2). Considering the Th1-related autoimmune particularities of NOD mice [62] and the absence of *Leishmania* data for this strain, we compared the *in vivo* parasite responses between the two strains in the presence and absence of OPN.

The parasite load and ear lesions showed few differences between these two strains (Fig 4A and 4B). In wild-type mice, the parasite content and ear width were lower in the NOD mice than in the C57BL/6 mice, especially between 16- and 70-days post-infection (expansion phase) (Fig 4A and 4B), while in the plateau phase (D91-121), the ear lesions were more pronounced in the NOD mice (Fig 4B).

In contrast, in the *opn* knockout mice, the parasitic load was consistently higher and similar between the two strains except in the ExP phase, whereas NOD$^{-/-}$ mice exhibited a higher parasitic load (Fig 4C). Additionally, the ear lesions of the NOD KO mice were less developed than those of the C57BL/6 KO mice (Fig 4D). These data indicate that the Th1-related immune nature of the NOD genome facilitates the proliferation of the parasites in the absence of *opn*, at

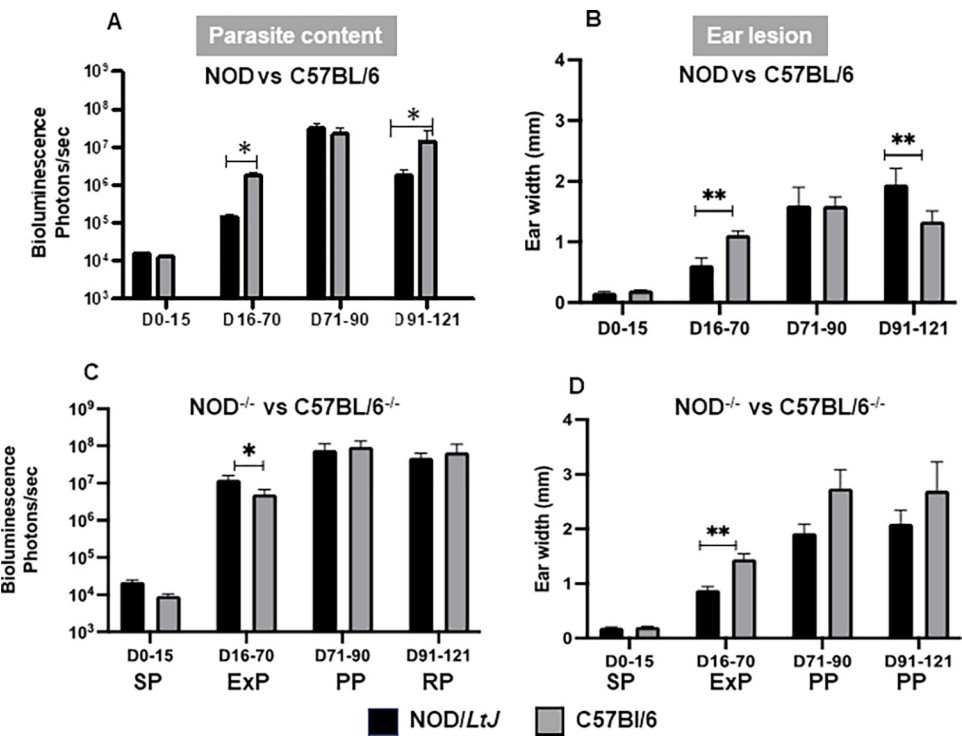

**Fig 4. Comparative analysis of *Leishmania amazonensis* infection between NOD and C57BL/6 mice in the presence or absence of OPN.** The cumulative data of the four phenotypic windows corresponding to days post-infection are presented. (A) and (C). Matched evaluation of the fluctuations in the parasitic load of the NOD/*LtJ* (black bars) and C57BL/6 (greygrey bars) strains by bioluminescence signalling (expressed in photons/sec/ear) at inoculation sites *in vivo*. (A): Wild-type, D16-70 (*P = 0.0469); D91-121 (*P = 0.0382) and **(C):** *Opn* knockout mice, D0-15, (***P = 0,0001) and D16-70 (*P = 0.0196). (B) and (D) Ear lesions (ear widths are expressed in mm). (B) Wild-type NOD and C57BL/6 D16-70 (*P = 0,0196); D91-121 (**P = 0.0028) and (D) *Opn* knockout animals; D16-70 (**P = 0.0005). A Mann–Whitney test was performed to compare the fold changes between the two groups. Nonparametric correlation tests (Spearman) were performed to compare the bioluminescence data.

least in the early phases (Fig 4C), while in its presence (Fig 4A), these responses are likely moderated by the synergistic effects exerted on the *opn* by the infection and the autoimmune-prone genetic setting in relation to the Th1/Th2 balance.

**Host cellular responses to *L. amazonensis*: *In vitro* infection of BMFs isolated from wild-type and *opn*-mutant NOD mice.** *L. am*. amastigotes are predominantly found within resident dermal macrophages [63] and in dendritic cells [64] in the mammalian host. We evaluated the response to infection by *L. am*. parasites and the implication of OPN in BMFs isolated from NOD mice.

We first evaluated the replication of parasites at the amastigote stage (the intracellular stage) in the BMFs at 24 h and 48 h *p.i.* in the presence and absence of OPN (Fig 5).

Immunostaining of BMFs isolated from NOD wild-type mice infected with parasites revealed increased proliferation of *Leishmania* amastigotes at 48 h compared with 24 h *p.i.* (Fig 5A). *Opn* KO macrophages were similarly infected by the parasites (Fig 5A); however, no or only minor proliferation was observed between 24 h and 48 h *p.i.* in the absence of OPN (Fig 5A). qRT–PCR of the *ssrRNA* of *Leishmania* confirmed these data (Fig 5B). Thus, in the NOD genetic background, osteopontin seems to participate in parasitic proliferation (Fig 5B and S3 Table).

Parasite burden evaluation (mean intensity and crowding) of *Leishmania* followed the same patterns observed by immunostaining and qRT-PCR and confirmed these data (Fig 5C and S3 Table). The mean intensity and degree of parasite crowding were significantly higher ($P<0.05$) in the wild-type mice at 48 h *p.i.* (12.13 and 24.07 parasites/cell, respectively) than in the KO mice (8.5 and 8.76 parasites/cell, respectively) (Fig 5C and S3 Table). Contrary to the NOD, in the BMF isolated from C57BL/6 *opn* knockout mice, higher infection rates were observed in comparison with those in wild-type cells, indicating that OPN is involved in cell protective responses against parasites in this strain [35]. However, while 100% of the NOD KO macrophages were infected with parasites at 24 h, the wild-type macrophages reached 90% cell infection only at 48h *p.i.* (S4 Fig).

These data are consistent with the implication of OPN in the adaptation of *Leishmania* parasites in their cellular niche, the BMF of NOD mice, possibly participating in Th1 parasite-elicited immune responses.

**L. amazonensis parasites stimulate *opn* gene expression and OPN protein in BMFs isolated from NOD mice.** The ability of parasites to multiply in the NOD BMFs in the presence of OPN rather than in its absence indicates that this protein is required at least for their proliferation in host macrophages. The effect of infection by *Leishmania* amastigotes on osteopontin gene expression was evaluated by immunostaining and qRT-PCR (Fig 6A and 6B, respectively). A four-fold increase of *opn* transcript levels in the presence of parasites relative to noninfected macrophages was observed at 48 h *p.i.* (Fig 6B). The density of the OPN protein was evaluated in the infected cells and compared with that in the noninfected control cells (Fig 6C). The OPN protein followed similar patterns at 48 h *p.i.*; however, at 24 h *p.i.*, an increase in the protein was also observed (Fig 6C).

These data, together with the lack of parasite proliferation in the NOD.*opn*$^{-/-}$ mice described above (Figs 5B and 6B), confirm that the endogenous presence of osteopontin in the NOD BMFs is involved in the host response to parasite infection. The Th1 immune environment of NOD mice and the Th1 cytokine properties of OPN seem to favour *Leishmania* parasites within the NOD BMF niche, while in contrast, the presence of OPN confers protection against early T1D onset only in the absence of infection (Fig 1A).

This effect seems to be OPN-dependent since the expression of CD44 transcripts, encoding the OPN receptor, remained unchanged in both wild-type and *opn* mutant mice (S5 Fig), suggesting a receptor-independent effect of the OPN molecule. Therefore, OPN favours parasite

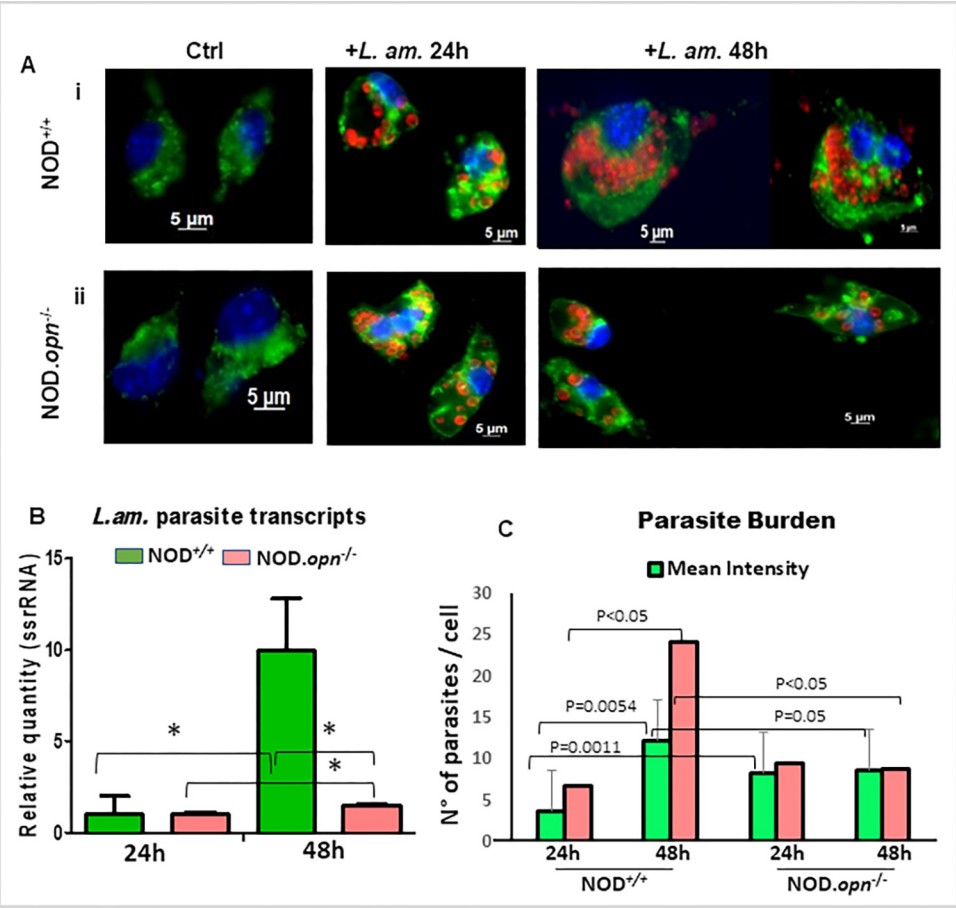

**Fig 5.** ***L. amazonensis* proliferation in BMFs in the presence or absence of osteopontin in the NOD genetic background.** (A) BMFs isolated from NOD/*LtJ* wild-type **(i)** or *opn* mutant (NOD.*opn*⁻/⁻ mice) **(ii)** were infected with *L. amazonensis* amastigotes at a ratio of 4:1. Representative images of BMFs populations and control non-infected BMFs (ctrl) are shown. Twenty-four and 48 hours later, each BMF population was analysed by immunostaining. Nuclei were stained with Hoechst (blue), vacuoles with Lysosome-associated membrane protein Lamp-1 Ab and FITC-labelled conjugate (green), and amastigotes with 2A3-A26 Ab and Texas Red-conjugate (red). All the images were acquired via phase-contrast optical microscopy. (B) Parasite quantification by qRT–PCR of the relative expression of the ssrRNA of *Leishmania amazonensis* in the cells in the presence or absence of OPN, as evaluated at 24 h and 48 h post-infection. Statistics: Unpaired t-test with Welch's correction: WT 24 h vs WT 48 h: P = 0.0364; WT 48 h vs KO 48 h: P = 0,0363; KO 24 h vs KO 48 h: P value = 0,0155. C. Parasite Burden in BMF. The mean intensities and intracellular parasite crowding of *L. amazonensis* amastigotes per infected cell were monitored by manual analysis of immunofluorescence image captures (AxioVision) of at least 2 different experiments (average number of 60 evaluated cells per condition). Statistical analyses were performed using the QP3.0 program designed for quantitative parasitology as described in the Methods section. The mean intensities were compared by the bootstrap test, and 2-sided bootstrap p-values are given as follows: WT vs KO at 24 h, P = 0.0011; WT 24 h vs 48 h, P = 0.0010; and KO 24 h vs 48 h, NS. Mean crowding was significant for the WT at 24 h vs. 48 h (P<0.05), the Cl at 97.5%, and the WT at 48 h vs. the KO at 48 h (P<0.05), and the Cl at 97.5%). The statistics are presented in S3 Table.

proliferation in the BMF *in vitro* (Fig 5), while *in vivo*, the presence of OPN seems to be host protective against *Leishmania* parasites, as shown by the clinical phenotype (Fig 2C) and parasite content in the infected tissue (Fig 2C and 2D). Different mechanisms and/or molecules implicated *in vitro* in the BMF and *in vivo* in the infected tissue may explain these observations.

The *L. am*. effect on the *opn* transcripts (Fig 6B) was compared between the NOD, DBA/2 and BALB/c mice (S6 Fig). Notably, BALB/c mice are sensitive to *Leishmania* parasites, while DBA/2 mice are resistant [65]. Interestingly, *opn* transcription was induced by the presence of

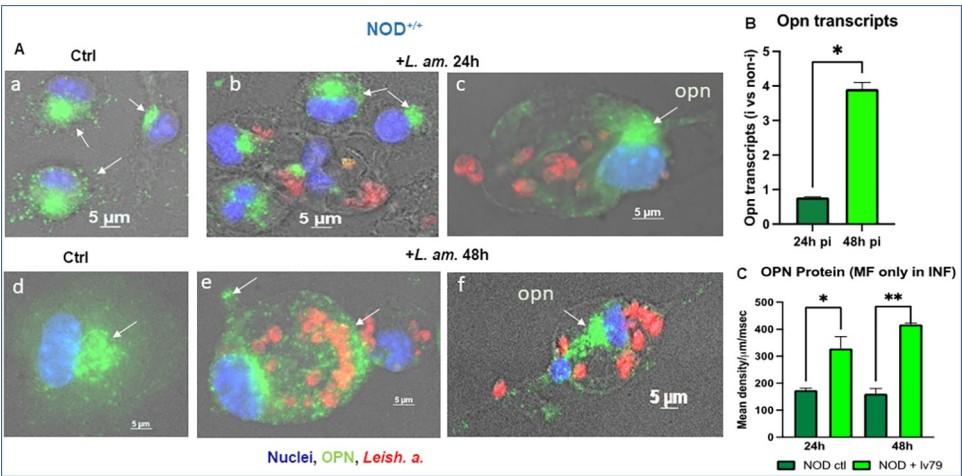

**Fig 6. OPN expression in the BMF of NOD$^{+/+}$ mice in the presence of *Leishmania amazonensis*.** (A) Immunofluorescence staining of BMFs. a & d: non-infected control at 24 h and 48 h, respectively. b & c: BMFs infected with LV79 at 24 h and **e** & **f** at 48 h post-infection. Green: OPN, red: LV79; blue: nuclei. (B) qRT–PCR of OPN transcripts (*P = 0.0294). (C) OPN protein quantification in control non-infected (ctl) and infected +LV79 BMFs. NOD ctl 24 h vs NOD+lv79 24 h (*P = 0.0454); NOD ctl 48 h vs NOD+lv79 48 h (**P = 0.0091). Two-tailed Mann–Whitney test.

parasites in BALB/c macrophages, while NOD mice showed lower *opn* gene expression and DBA/2 intermediate values at 24 h *p.i.* Therefore, *Leishmania*, in a permissive sensitive genetic background (BALB/c mice), stimulates early *opn* gene expression (24 h *p.i.*), which in turn facilitates parasitic proliferation in BMFs (S6 Fig). Overall, the variation in *opn* gene expression in response to *L. am.* observed between these three strains outlines i) the role of OPN in parasite proliferation independently to the genetic susceptibility to leishmaniasis disease and ii) a strain-specific response to the parasites requiring OPN, together with possible additional molecules participating in the variation in the host response to the parasites.

**OPN favours parasite proliferation in macrophages in the NOD genetic background.** To further evaluate the response of NOD mice to parasitic infection, we assessed the role of OPN in cell infectivity and survival by immunostaining BMF cells infected with *Leishmania* parasites at 24 h and 48 h *p.i.* in the presence or absence of OPN (Fig 7 and S7 Fig). OPN is expressed in both intracellular (iOPN) and extracellular (sOPN) forms in the BMF (Fig 7A, i & ii). High parasite proliferation was observed in the macrophages at 48 h *p.i.* in the presence of OPN (Fig 7B vii and S6ii, iv, v, Fig). The cells showed intact, non-apoptotic nuclei (Fig 7 and S7i and ii Fig). In contrast, the number of parasites was lower in the NOD *opn* knockout mice (S7 Fig iii and vii). We identified a similar phenotype in *Leishmania*-infected BMFs isolated from C57BL/6 mice, but only in the absence of OPN [35]. According to these data, the inflammatory response to the parasites is contained as shown by the modulation of *il-1β* transcripts at 48 h *p.i.* in the presence of OPN as well as in its absence (Table 1). However, the two-fold increase of *il-1β* transcripts at 24 h *p.i.* in NOD wild-type mice indicates an initial host response to the parasites, also dependent on the presence of OPN, as confirmed by the down-regulation of *il-1β* transcripts in its absence (Table 1). Similar results were obtained for the ear pinna after *L. am.* infection *in vivo* (Fig 8). Il-1β is a key mediator of the inflammatory response to pathogens [66]. *Leishmania spp.* parasites effectively down-regulate *il-1β* in C57BL/6 mice [35]. The increase in *il-1β* transcripts on day 100 *p.i.* may indicate the presence of a local reservoir of parasites after infection, as previously reported for *L. major* [67]. Together, these observations suggest that i) in the macrophages isolated from the NOD mice,

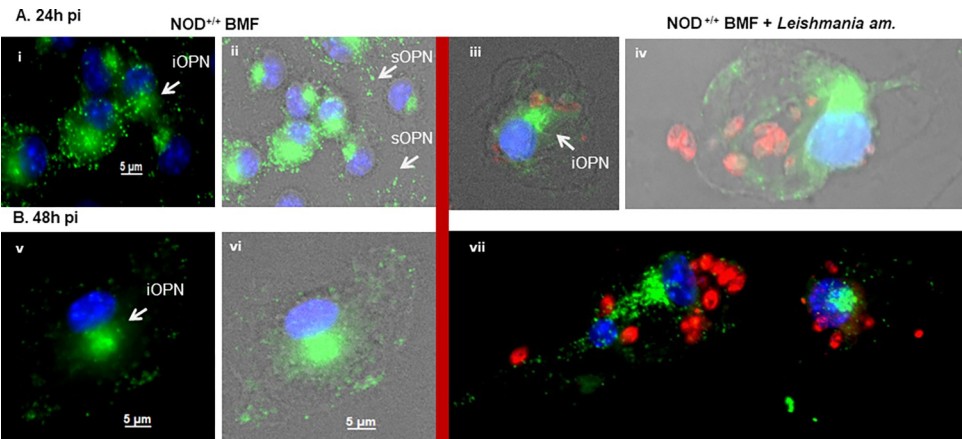

**Fig 7. Infectivity and survival of *Leishmania amazonensis* parasites in BMFs isolated from NOD mice.**
Representative images of immunofluorescence staining for OPN protein (green) and *Leishmania* parasites (red); nuclei
are blue. (A) BMFs at 24 h in non-infected (i and ii) and infected (iii and iv) cultures. (B) BMFs at 48 h in non-infected
(v and vi) and infected (vii) cultures. Total N° of cells at 24 h: WT: 117; KO: 29 and at 48 h *p.i.* WT: 55; KO: 4. iOPN:
intracellular OPN; sOPN: secreted OPN. Images were acquired with a Zeiss AxioVision Rel. 4.8.2 Image acquisition
software (Carl Zeiss International).

OPN favours the growth of the amastigotes and ii) the parasites while eliciting the host
response, as shown by the increase in *il-1β* transcripts at 24 h *p.i.*, confer an adaptation to the
host, as indicated by the subsequent downregulation of *il-1β* at 48 h *p.i.* Overall, the NOD
genetic background seems to represent a parasite-favourable cellular environmental niche.

## Immune host response to *L. amazonensis* and the role of OPN

***L. amazonensis* stimulates iNOS expression via an OPN-dependent mechanism.**
Microbial compounds and LPS activate NO in macrophages through the enzymatic action of

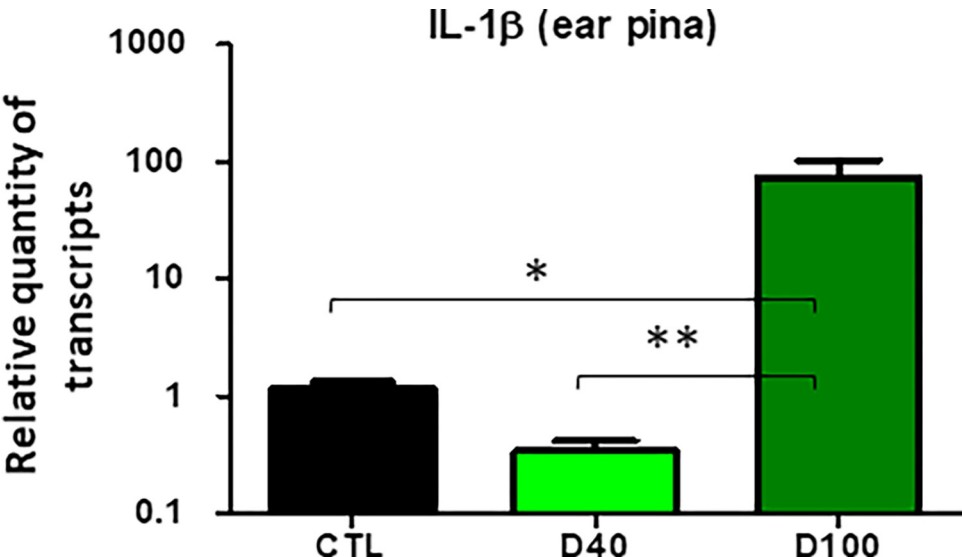

**Fig 8. *In vivo Il-1β* transcripts.** qRT–PCR of Total RNA isolated from infected ear pinna tissue from WT (CTL)-
infected, $D_{40}$ *p.i* and D40 KO mice at $D_{40}$ and $D_{100}$ *p.i.* (Ctl vs $D_{100}$ *p.i.*: P = 0,0051** and KO $D_{40}$ vs KO $D_{100}$ *p.i.*:
P<0.0001). F-test to compare variances.

**Table 1. Host response to *Leishmania amazonensis* and OPN role in immune responses. *In vitro.*** Transcript profiles encoding for IL-1β and for iNOS, STAT-1, IL-17 (Th1) and IL-10, IL-4 (Th2) cytokines in BMF isolated from NOD[+/+] (WT) and NOD.*opn*[-/-] (KO) mice in response to *L. amazonensis*. Cells were inoculated with *Leishmania amazonensis* amastigotes at a MOI: 4:1 (parasites: macrophages) and incubated for 24h and 48h before lysis for RNA preparation. Mean fold changes for transcripts and standard deviations have been calculated by using as a calibrator the wild-type non-infected value at each time point.

| Transcripts | BMF | 24h *p.i.* | 48h *p.i.* |
|---|---|---|---|
| **IL-1β** | WT | 2.17 ± 0.2 | -1.42 ± 0.18 |
| | KO | -1.2 ± 0.06 | -1.47± 0.05 |
| **iNOS** | WT | 64.44 ± 2.475 | 185.3 ± 3.182 |
| | KO | 32.50 ± 1.768 | 30.50 ± 1.768 |
| **STAT1** | WT | 6.077 ± 0.01453 | 14.09 ± 0.2991 |
| | KO | 2.977 ± 0.5488 | 1.593 ± 0.1386 |
| **IL-17** | WT | -15.22 ± 0.5691 | -149.5 ± 0.9780 |
| | KO | 18.46 ± 0.1125 | 20.97 ± 0.5000 |
| **IL-4** | WT | -1.9 | -2.6 |
| | KO | -2.55 | -2.62 |
| **IL-10** | WT | 0.7700 ± 0.02 | 12.35 ± 0.44 |
| | KO | 21.15 ± 0.16 0.7700 | 56.16 ± 0.17 |

inducible nitric oxide synthase (iNOS), which represents one of the major microbicidal mechanisms of macrophages against pathogens [68–70]. The antimicrobial role of NO in the control of infectious parasites has been demonstrated in macrophages [71–74], including in the control of *Leishmania* proliferation [75]. Regulation of the inhibition of *Leishmania* parasite growth is mediated by the feedback of NO production by iNOS through the action of OPN.

Osteopontin, induced by NO, down-regulates iNOS via an autocrine mechanism upon stimulation of the ubiquitin (Ub)-proteasome degradation factor, which inhibits signal transducer and activator of transcription 1 (STAT1), the critical transcription factor for iNOS gene expression [76]. However, *Leishmania* parasites can also act directly on stat1. It was demonstrated that *L. major* and *L. mexicana* attenuate stat1 and *L. donovani* infection of macrophages attenuates tyrosine phosphorylation of JAK1, JAK2 and stat1, subsequently blocking IFN-γ and thus evading host defence mechanisms [77,78].

We addressed the effect of the parasites on the iNOS transcripts in the BMF in the presence or absence of OPN (Table 1). In the BMF of NOD wild mice, iNOS transcripts were highly upregulated in comparison to the non-infected cells, reaching 64-fold at 24 h *p.i.* and 185-fold after 48 h *p.i.* (Table 1). This up-regulation was inhibited in the absence of OPN (Table 1).

We addressed the transcript levels of *stat1*, known to stimulate iNOS gene transcription. In contrast, *stat1* transcript levels increased 6- and 14-fold (at 24 h and 48 h *p.i.*, respectively) in the wild-type BMFs, while low levels were detected in the KO cells, indicating the importance of OPN (Table 1). Stat1 is known to act as a dimer after phosphorylation and nuclear translocation to trigger the transcription of its targets [79]. The activation of *stat1* in the BMF (14-fold increase at 48 h, Table 1) correlated with the increase in iNOS. However, the parasite burden was higher in NOD wild-type mice despite high iNOS gene expression (Fig 5B), while in the absence of OPN, the parasite burden was lower despite the decrease in iNOS transcripts, indicating the necessity of the presence of OPN for parasite proliferation (Fig 5C). The presence of iNOS together with the sustained presence of the parasites in the NOD BMFs is unexpected unless another mechanism is involved containing the host response to the parasites and related to the presence of OPN.

Therefore, despite the known effect of iNOS on pathogen control, it seems that *Leishmania* parasites have evolved additional complex adaptation mechanisms to dampen host defence responses. Two hypotheses may be proposed for these observations: i) in NOD mice, NO does

**Table 2. Host response to *Leishmania amazonensis* and OPN role in immune responses. *In vivo*.** Transcript profiles as indicated in the ear pinna and draining lymph nodes (DLN) of NOD$^{+/+}$ (WT) and NOD.*opn*$^{-/-}$ (KO) mice in response to *L. amazonensis*. $10^4$ metacyclic promastigotes of parasites (LV79 strain) were inoculated in the ear dermis of the mice and transcript profiles were evaluated in the temporal windows spanning the early and expansion phase (EP-ExP) at Day$_{40}$ *p.i.* and the parasite reduction phase (RP) at Day$_{100}$ *p.i.* (see Fig 2). Mean fold changes for transcripts and standard deviations have been calculated in the *opn* KO mice by using as a calibrator the wild-type D$_{40}$ *p.i.* value. Significant differences (p<0.05*) are indicated (Mann-Whitney test or Kruskal Wallis test).

| Transcripts | Tissue | ExP-D$_{40}$ (KO/WT) M +/- SD | RP-D$_{100}$ (KO/WT) M +/- SD |
|---|---|---|---|
| iNOS | Ear | 0.42+/-0.073 * | 46.83+/-33.28 * |
|  | DLN | 0.9167 ± 0.2437 | 13.50 ± 12.20 |
| STAT1 | Ear | 0.79+/-0.1302 | 2.14+/-0.95 |
|  | DLN | 0.72+/-0.38 | 0.41+/-0.06 |
| IL-17 | Ear | 0.013 ± 0.0033 | 104.5 ± 100.2 |
|  | DLN | 85.86 ± 78.72 | 1481.34+/-979.8 |
| IL-4 | Ear | 0.33+/-0.058 | 15.18 ± 14.59 |
|  | DLN | 3.45+/-3.5 | 30.79+/-25.1 |
| IL-10 | Ear | 2.533 ± 0.9543 | 578.1 ± 547.8 |
|  | DLN | 1.003+/-0.44 | 0.66+/-0.05 |

not act directly to contain the parasites; other mechanisms seem to be involved; and ii) an insufficient presence of oxygen may dampen NO production despite high iNOS, resulting in no clearance of the parasites. A similar phenomenon was demonstrated in macrophages infected with *L. major*, whereas the absence of oxygen prevented the production of enough NO to clear *L. major* [80].

The downregulation of iNOS *in vivo*, at the expansion phase (ExP) of the parasite (D$_{40}$), at the sites of inoculation (Table 2) and in the draining lymph nodes (DLN) (Table 2) of the *opn* knockout mice, as well as the increase in iNOS, observed at D$_{100}$ *p.i.* (reduction (RP) phase), correlates with the low *stat1* expression (Table 2) and may indicate parasite persistence in tissue reservoirs, as previously demonstrated [81]. Additional studies are required to confirm and elucidate these observations.

**Role of OPN in the Th1-Th2 immune balance of the host response to *L. amazonensis*.** Osteopontin, as mentioned above, is a Th1 cytokine expressed in cells of the adaptive (T cells) and innate (macrophages, DCs) immune system [82]. Using qRT-PCR, we evaluated the transcripts encoding genes that promote Th1 or Th2 host immune responses to *L. am.* parasites in the NOD mice in the absence or presence of osteopontin *in vitro* in the BMF (Table 1). *Leishmania* infection strongly inhibited interleukin-17 (*il-17*) transcript levels in the NOD BMFs (Table 1). Similar results were obtained for the *il-4* transcripts (Table 1). However, in the absence of OPN, *il-17* was upregulated approximately 20-fold (Table 1), while *il-4* remained downregulated (Table 1). In *Leishmania*-susceptible BALB/c mice, *il-4* was reported to be elevated after infection, which leads to progressive disease [83, 84]. The downregulation of *il-4* in the NOD mice indicates that this strain is resistant to the parasites.

In contrast, *il-10* transcripts were induced by the parasites in the presence of OPN but only after 48 h *p.i.* (12-fold) (Table 1). Interestingly, *il-10* transcripts were 50-fold over-induced at 48 h *p.i.* by the parasites in the BMF of the *opn*-knockout mice (Table 1). Similar *in vivo* data were obtained for the infected ear pinna of the NOD *opn* mutant mice, but only at 100 days *p.i.* (Table 2). These data indicate that in NOD mice, the Th1 inner immune proinflammatory environment is contained by the parasites, while in the absence of OPN, the increase of *il-10* transcript level (Table 2) indicates a counterbalance towards a protective for the parasites' host environment and parasite proliferation. The increase of *il-10*, a Th2-related cytokine, in macrophages (Table 1) correlated with the inhibition of parasite proliferation in the presence of OPN *in vivo* (Fig 2B and 2D). Il-17 was shown to promote the progression of cutaneous

leishmaniasis (CL) in BALB/c mice, which are known to develop Th2 immunity and succumb to infection [85]. In contrast, in NOD mice, which are known to possess Th1 immunity, *il-17* is inhibited by parasites in the BMFs (Table 1). Increased levels of *il-17*, as well as *il-10* in the absence of OPN both *in vitro* and *in vivo*, seem to be under the control of infection by the parasites and correlate with the post-infection acceleration of T1D (Fig 2B). Indeed, NOD mice deficient in il-10 were protected from organ-specific autoimmunity [86]; therefore, in our experimental setting, the increase in *il-17* and *il-10* transcript levels in *opn*-deficient mice correlated with the T1D acceleration observed after infection by the parasites. Overall, these data are consistent with the increased proliferation of the parasites in the wild-type BMF (Fig 5), indicating host adaptation of the parasites in the presence of OPN.

**Low expression of IFN-γ in the absence of OPN infers to a Th2 shift promoted by *L. amazonensis* infection.**   The presence of IFN-γ in Th1 immune cells and the role of this cytokine in preventing the shift to Th2 immunity [87, 88] prompted us to assess its presence in the ear lesions of infected NOD mice. In the presence of the parasites, IFN-γ transcript levels were low in the ear lesions of both wild-type and KO mice (Table 3), while in the DLN, a fivefold increase was observed in the presence of opn (Table 3), indicating that IFN-γ-producing cells were present in the DLN after infection with the parasites. The ratio of *IFN-γ*: *il-10* remained low in the absence of OPN at the sites of parasite inoculation (Table 3), while the *IFN-γ*: *il-4* ratio increased 3-fold (Table 3). However, on $Day_{100}$ *p.i.*, both ratios (*IFN-γ*: *il-10* and *IFN-γ*: *il-4*) remained low, indicating a shift towards a Th2 response in the absence of OPN and correlating with the high levels of *il-4* and *il-10* observed in the ear lesions as well as in the DLN (Table 2). These data are consistent with published observations showing that *IFN-γ* critically regulates *opn* gene expression. It was reported that in human monocytes, while OPN positively regulates *IFN-γ* expression, this cytokine in turn stimulates OPN expression in a positive regulatory loop during infection [89]. Therefore, in our study, in the absence of OPN, *IFN-γ* expression was low, and the parasites created a permissive Th2-dependent environment, especially at $D_{100}$ *p.i.*, whereas an increase in Th2 transcript levels was observed (*il-4* and *il-10*, Table 2).

The role of IFN-γ in *Leishmania* infections has long been debated. In the absence of IFN-γ, C57BL/6 mice are highly susceptible to *L. major* and exhibit an increase in Th2-type responses [90], while two months after infection with *L. am.* they develop late severe disease [90]. Similarly, the role of IFN-γ in T1D, ranging from no effect to disease exacerbation, is unclear [90, 91]. In recent work, the loss of IFN-γ signalling was shown to affect the incidence of T1D but not prevent it [92]. The acceleration of T1D in the absence of OPN after infection with the parasites and the proliferation of *L. am.* observed in the knockout mice is concordant with a shift towards a Th2 phenotype resulting in exacerbation of the clinical *Leishmania* phenotype (Fig 3). These observations are in agreement with the low levels of *IFN-γ* observed in the absence of OPN, while the parasites delayed T1D development (Fig 1) and aggravated the inflammatory clinical phenotype (Fig 2).

**Table 3. Host response to *Leishmania amazonensis* and OPN role in immune responses.** Th2 parasite-dependent shift delineated by low IFNγ expression in the absence of OPN.

| Transcripts | Tissue | NOD/LtJ (WT) | ExP-$D_{40}$ (KO/WT) | RP-$D_{100}$ (KO/WT) |
|---|---|---|---|---|
| **IFNγ** | Ear | 1.100 ± 0.1004 | 1.017 ± 0.2315 | 0.004759 ± 0.0041 |
| | DLN | 5.220 ± 0.2268 | 2.704 ± 0.8388 | 2.4 ± 0.26 |
| **IFNγ/IL-4** | Ear | 1.100 ± 0.1004 | 3.23 ± 0.93 | 0.0132 ± 0.01 |
| **IFNγ/IL-10** | Ear | 1.100 ± 0.1004 | 0.5 ± 0.09 | 0.00009 ± 0.000066 |

In the absence of OPN, the increase in the *IFN-γ*: *il-4* ratio in the ear lesions of the *opn* knockout animals *in vivo* at $D_{40}$ *p.i.* (Table 3) indicates that a Th1/Th2 balance is occurring, probably related to a wound repair mechanism that is dependent upon the presence of macrophages and other lymphocytes producing IFN-γ [93]. This suggests rudimentary conservation of Th1 immunity after infection with *Leishmania* parasites, even in the absence of OPN. However, further analysis is necessary to confirm these data and the proposed hypothesis.

## Discussion

Evolutionary constraints imposed by infectious microorganisms on their hosts impact the adaptation of both the microorganisms and their mammalian hosts. These environmentally imposed cohabitations led to mutualism, commensalism, or parasitism. In the case of parasites, the remnants of such adaptation are present in both mammalian hosts and parasites. The adoption of macrophages as a niche for *Leishmania* parasites is one remarkable example of this process, as both the host cell and the parasite have found mechanisms for coexistence.

Genetic changes have been fixed throughout evolution conferring adaptation possibilities between parasites and their hosts. The epigenetic influence of parasites on their host also may have been permissive on specific genes that have been part of host-parasite adaptation. The "Hygiene" hypothesis includes the possibility that such genetic changes may be etiological triggers of or protective against autoimmunity. For example, the control of CL, as revealed by genetic studies, depicted a complex picture dependent upon the genetic background of the animals under study. Genetic loci influence the host's response to parasites [94, 95]. H2 loci were found to influence the outcome of infection, and depending on the HLA alleles, mice are classified as resistant or prone to CL [96–98]. Similarly, HLA loci, particularly *HLA*-DR/DQ haplotypes, are associated with autoimmune diabetes [99]. According to the H2 haplotypes, NOD mice predisposed to T1D carrying the $H2^{g7}$ $K^d$ alleles may be classified as resistant to *Leishmania*, yet they share other *Leishmania* susceptibility loci within chromosomal regions of T1D susceptibility [100–102].

Subversion of macrophages as host cells for *L. am.* parasites is essential for established and persistent infection by parasites. We have previously demonstrated that during *L. am.* infection in C57BL/6 mice, OPN is implicated in the strong inhibitory effect of the parasite on the inflammatory response [35]. In the absence of OPN, this effect was moderate, indicating that OPN is part of the host's response to the parasites. Infection with *L. am.* of the NOD mice showed an acceleration of T1D onset in the presence of OPN, while in its absence, T1D decreased and remained lower than that in the non-infected animals (Fig 1B). Therefore, the delay of T1D observed in the absence of OPN after parasite infection indicates that at least in part, OPN-dependent immune mechanisms, known to participate in autoimmune disease in NOD mice, are elicited by the parasites [103]. Thus, the protective effect of OPN against T1D was partially abolished by *Leishmania* infection (Fig 1B). Moreover, OPN is known to modulate the host response to infection and to participate in immune-mediated inflammation through Th1 cytokine activity in cell-mediated immunity [104, 105]. While Th1 responses are known to be important for parasite clearance in mice, Th2 responses (*i.e.*, IL-4) are associated with parasite persistence and disease progression [106, 107]. However, human leishmaniasis exhibits mixed Th1/Th2 responses, indicating a more complex immune response in humans infected with cutaneous *Leishmania* species than in mice [108].

Interestingly, in NOD mice, *Leishmania* infection strongly inhibited *il-17* transcripts (149.5-fold down-regulated at 48 h *p.i.*, Table 1) in the BMF, potentially facilitating parasite survival. Indeed, a low *il-17* response corresponds to host resistance to *Leishmania* [85]. In contrast, in the absence of opn, the *il-17* transcript level increased, indicating that opn plays a

role in the suppression of *il-17* in the presence of parasites and thus participates in protection against *Leishmania* in NOD mice. Il-17 has been implicated in T1D regulation in mice, particularly in the later effector phase of the disease [109, 110], and recent reports indicate that Il-17 is part of the aetiology of T1D [111]. Further research may elucidate the precise mechanisms underlying the interplay between opn, il-17 and T1D in the context of *Leishmania* infection.

Another interesting observation of our data is the variation in the transcripts encoding il-10. In our experimental setting, infection with *L. am.* increases *il-10* transcripts in the BMFs of both NOD wild-type and *opn* knockout mice (12-fold and over 50-fold, respectively; Table 1). Il-10 is an anti-inflammatory cytokine that strongly suppresses Th1-type immune responses [112]. It is involved in the Th1/Th2 immune balance in T1D, while it plays an important role in infections [86, 113]. The absence of il-10 has also been reported to exacerbate both innate and adaptive immunity in response to *Listeria monocytogenes* [114] and plays a role in several other infections [115]. Moreover, IL-10 deficiency exacerbates T-cell-mediated autoimmune diseases [86, 113, 116, 117]. In T1D, differential regulation of il-10 has been reported, and a role for this gene in the modulation of both innate and adaptive immune cells and the development of autoimmunity has been proposed [113]. This finding suggested the involvement of OPN in modulating the expression of *il-10* and possibly influencing the immune Th1/Th2 balance. These data corroborate the role of OPN in the efficient development of Th1 immune responses [105].

We assessed the expression of *il-4*, a Th2 cytokine recognized for its role in susceptibility to intracellular pathogens when its expression is markedly elevated. It was shown that suppression of il-4 production during early *L. major* infection prevents the differentiation of Th2 responses [118]. It has also been reported that il-4 is involved in the initiation of Th1 in BALB/c mice in response to *L. major*, leading to resistance to the disease [119]. Although the role of il-4 in Th2 differentiation has been well described, il-4 may also have opposite effects on Th2 immunity [120]. Interestingly, in NOD BMFs infected with *L. am. il-4* transcripts were inhibited independently of the presence of OPN (Table 1), suggesting that other proinflammatory cytokines known to be potentiated by Il-4 may also be absent or inhibited by the parasites in these cells. These data are consistent with the increased proliferation of the parasites in the wild-type BMFs (Fig 5), indicating an OPN-dependent adaptation to the parasites.

OPN in a non-infectious environment is known to repress iNOS expression by increasing the ubiquitination and degradation of stat1, the transcription factor for iNOS gene expression [76, 121]. However, the regulation of iNOS expression is complex and contingent upon diverse stimuli, the local cytokine content, and prevailing pathophysiological conditions. One mode of iNOS regulation involves the activation of its promoter by a list of transcription factors, dictating a context-dependent outcome, wherein iNOS expression may be either activated or restrained based on the specific cytokine and microbial stimuli, as well as the cellular context [122, 123]. The data presented revealed a pattern wherein both iNOS and *stat1* are upregulated in the presence of OPN, while when OPN is absent, both transcripts decrease (Table 1), indicating that OPN plays a role in the upregulation of these genes in this context. Therefore, infection by parasites activates the iNOS/OPN/STAT1 pathway, suggesting that parasite-induced iNOS activation might contribute to parasite persistence in macrophages, potentially through the inhibition of the NLRP3 inflammasome by iNOS [124] and our unpublished observations). Additionally, in the absence of OPN, the low expression of iNOS and *stat1* transcripts, along with the increase in *il-10* transcripts, indicates a major role for OPN in the Th1/Th2 balance in NOD mice infected with *L. am*.

Overall, the key findings include the following: i) Differential effects of *L. am.* infection on the T1D phenotype between wild-type and knockout mice: while wild-type mice show increased T1D incidence after infection, in the absence of OPN, infection with the parasites

decreases the incidence of the disease; ii) a higher parasitic load is observed in infected wild-type BMF at 48 h *p.i.*, while no parasitic proliferation is observed in the NOD *opn* null mice; iii) discrete variations in the *in vivo* clinical phenotype as well as in parasite content at the infection sites are notable in the absence of OPN; and iv) *p.i.* Finally, Th1 responses elicited by the parasites in NOD BMF were abolished in the absence of OPN, while Th2 responses increased.

These findings underscore the intricate interplay between OPN, immune responses, and disease outcomes in the context of both T1D and leishmaniasis, suggesting that OPN is a potential target for therapeutic interventions.

## Conclusions

In conclusion, the influence of microorganisms on the development of autoimmune responses poses a complex and intriguing question. Our study investigating the impact of *L. amazonensis* parasites on type 1 diabetes provides insights into potential immune mechanisms at play in autoimmune-prone NOD mice within an infectious context.

This work serves as an exploratory investigation, and further studies are needed to confirm and expand upon our findings. Additionally, exploring the involvement of genes shared between *Leishmania* and T1D susceptibility loci could offer valuable insights into the interactions between the host and parasites in an autoimmune genetic context.

In this context, our findings demonstrate that a) environmental triggers, such as infection by the parasite *L. amazonensis*, which elicits Th1 immune responses similar to those observed in NOD mice, can impact autoimmunity and b) interactions between the host and parasites may shape singular associations that influence natural selection.

Overall, our study highlights the intricate interplay between environmental factors, immune responses, and genetic predisposition in the context of autoimmune diseases. Further research in this area will contribute to a better understanding of autoimmune pathogenesis and may offer new avenues for therapeutic interventions.

## Supporting information

**S1 Table. Sequences of primers used for qRT-PCR.**
(PDF)

**S2 Table. Type 1 diabetes in response to *Leishmania amazonensis*.**
(PDF)

**S3 Table. Comparative statistical evaluation of parasite infectivity in BMF.**
(PDF)

**S1 Fig. FACS analysis of Bone Marrow precursor-derived macrophages (BMF) and Dendritic cells (DC). A.** BMF cell markers and **B.** Dendritic cell markers. **C.** Summary showing similar characteristics of the BMF and DC between the NOD/LtJ (or NOD$^{+/+}$) wild type and *opn* (NOD.*opn*$^{-/-}$) knockout mice. The DC population was MHC-II$^{low}$ and mainly CD11c$^-$ with few clusters of CD11c$^+$ containing cells.
(PDF)

**S2 Fig. Physiology of infection. A.** Mean weight (gr) of the NOD wild-type (NOD$^{+/+}$) and *opn* knockout (NOD.*opn*$^{-/-}$) mice after *L. amazonensis* infection. **B.** Post-infection variation of body weight, by the age of mice. **C.** Glycemia in infected with *L. amazonensis* NOD wild type and NOD *opn* knockout mice at 22 weeks of age.
(PDF)

**S3 Fig. Histological examination of the infected sites in the ear pinna. A & B.** Infected ear pinna from NOD wild-type mice (NOD$^{+/+}$). **C & D.** Infected ear pinna from NOD.$opn^{-/-}$ mice. Hematoxylin and Eosin (H&E), **A & C**: Original Magnification x4, scale: 250, **B & D**, Original Magnification x4, scale bar: 50 μm. E: epidermis, D: dermis, C: cartilage. Arrows show the accumulation of parasites in the inflammatory cells.
(PDF)

**S4 Fig. Efficiency of cell transmission of *Leishmania amazonensis* parasites in bone marrow-derived macrophages (BMF).** Representative experiment of BMF infected with *Leishmania amazonensis*, in the presence (NOD$^{+/+}$) or in the absence (NOD.$opn^{-/-}$) of osteopontin at 24h and 48h *p.i.* Total cell numbers are presented in S2 Table (QP 3,0 program for stats). NOD$^{+/+}$ at 24h: 152 cells, NOD$^{+/+}$ at 48h: 58 cells. NOD.$opn^{-/-}$ at 24h: 29 cells and for NOD$^{-/-}$ at 48h: 8 cells.
(PDF)

**S5 Fig. Real time PCR for the OPN receptor CD44 in the BMF.** Two sets of primers (CD44-1 and CD44-2) were used (see S3 Table).
(PDF)

**S6 Fig. Comparison of the relative quantities of *opn* transcripts by q-RT-PCR of NOD/*LtJ*, DBA/2 and Balb/c strains of mice.** Relative quantities are expressed in ratios of infected (i) versus non-infected (non-inf) *opn* transcripts in the BMF at 24h post-infection.
(PDF)

**S7 Fig. OPN favours parasite proliferation in the NOD genetic background (NOD$^{+/+}$).** BMF + Leishmania am. (LAMP Immunostaining). A. BMFs isolated from NOD wild-type (NOD/LtJ or NOD$^{+/+}$) mice at 24h p.i. with L. am. (i) and at 48h p.i. (ii). B. BMFs isolated from NOD KO (NOD.opn$^{-/-}$) mice at 24h *p.i.* (iii and iv) and at 48h p.i. (v, vi and vii). Pyroptosis-like cell swelling and membrane blending with intact nuclei and releasing of the parasites were observed only in the presence of OPN (ii). The numbers of cells examined are as in the legend of Fig 7. Parasite Numbers, crowding WT vs KO at 48h: 24.07 vs 8.76; P<0.05, CI 97.5%. Data and statistics are from the QP 3.0 program as described in S2 Table and M&M.
(PDF)

## Acknowledgments

We acknowledge the members of the Immunophysiology and Parasitism Unit who participated in the experimental part of this work.

## Author Contributions

**Conceptualization:** Evie Melanitou.

**Data curation:** Emilie Giraud, Laurence Fiette, Evie Melanitou.

**Formal analysis:** Emilie Giraud, Evie Melanitou.

**Funding acquisition:** Evie Melanitou.

**Investigation:** Laurence Fiette, Evie Melanitou.

**Methodology:** Emilie Giraud, Laurence Fiette, Evie Melanitou.

**Supervision:** Evie Melanitou.

**Validation:** Emilie Giraud, Evie Melanitou.

**Visualization:** Emilie Giraud, Laurence Fiette, Evie Melanitou.

**Writing – original draft:** Evie Melanitou.

**Writing – review & editing:** Emilie Giraud, Laurence Fiette.

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
