## [Decision Letter · Decision Letter 0]

1 Aug 2024

Type 1 diabetes and parasite infection: an exploratory study in the NOD mouse

PONE-D-24-24715

Dear Dr. Melanitou,

We’re pleased to inform you that your manuscript has been judged scientifically suitable for publication and will be formally accepted for publication once it meets all outstanding technical requirements.

Kind regards,

Nisha Singh, Ph.D.

Academic Editor

PLOS ONE

Additional Editor Comments:

I would like to clarify the rationale behind my decision to accept the manuscript "Type 1 diabetes and parasite infection: an exploratory study in the NOD mouse" despite Reviewer 2's recommendation for minor revisions.

1. After carefully considering the reviewer's comments and conducting a thorough evaluation of the manuscript, I determined that the suggested revisions were insignificant and did not impact the overall quality or validity of the study's findings.

2. The abstract is well-constructed and adheres to the principles of scientific writing. Additionally, acronyms are well defined in the manuscript, ensuring clarity for the readers.

3. The manuscript presents a significant contribution to the field and is well-written. The suggested revisions mainly pertain to minor language adjustments, which I believe is not an accurate assessment of the manuscript. The language of the manuscript is excellent.

4. While the reviewer mentioned issues with the quality of figures, the core content and data presented are clear and interpretable. Figures provided by the authors in the supporting files are of good quality, and any further improvements can be made during the final production stage.

5. The recommendation to include recent references is a valid point but does not detract from the overall contribution and quality of the manuscript. Since the research work is novel, it is not necessary that such related work has been done in the recent past.

6. The authors have addressed future directions, stating, "Further research in this area will contribute to a better understanding of autoimmune pathogenesis and may offer new avenues for therapeutic interventions," which is quite specific.

Considering these points, I concluded that the manuscript in its current form is suitable for publication and that the minor revisions suggested by the reviewer could be addressed during the final editing stages. This approach aims to expedite the dissemination of valuable research findings to the scientific community.

2. To comply with PLOS ONE submissions requirements, in your Methods section, please provide additional information regarding the experiments involving animals and ensure you have included details on methods of sacrifice, and efforts to alleviate suffering.

This work was supported by funding from the Institut Pasteur

Please respond by return e-mail so that we can amend your financial disclosure and competing interests on your behalf.

6. Please amend either the abstract on the online submission form (via Edit Submission) or the abstract in the manuscript so that they are identical.

Reviewers' comments:

Reviewer's Responses to Questions

**Comments to the Author**

1. Is the manuscript technically sound, and do the data support the conclusions?

Reviewer #1: Yes

Reviewer #2: Yes

2. Has the statistical analysis been performed appropriately and rigorously? 

Reviewer #1: Yes

Reviewer #2: Yes

3. Have the authors made all data underlying the findings in their manuscript fully available?

Reviewer #1: Yes

Reviewer #2: Yes

4. Is the manuscript presented in an intelligible fashion and written in standard English?

Reviewer #1: Yes

Reviewer #2: Yes

5. Review Comments to the Author

**Reviewer #1:** The manuscript seems compact and chiefly describes environmental infectious

component of the autoimmune response, including through Th1 immunity common in type I diabetes and leishmaniasis. The study encompasses novelty and approach to the experiments performed to prove the hypothesis seems quite satisfactory since the data of supplementary figures is provided.

**Reviewer #2:** The manuscript titled " Type 1 diabetes and parasite infection: an exploratory study in the NOD mouse” submitted in PlOS ONE journal is found it to be a well-structured and informative work. However, I have a few suggestions for minor revisions that I believe will further enhance the clarity and impact of your manuscript:

Abstract section is need to be updated. Avoid abbreviation in the abstract and arrange in one paragraph.

I suggest highlighting specific contributions or insights gained from the research in the abstract section. This will help differentiate your study from existing literature reviews on the topic and emphasize its novelty.

Avoid to use of personalized words (like we, I, ect) (Page 2, Line 45)

Arrange significant figure through the manuscript.

The Introduction section requires updating with recent relevant research. I recommend incorporating the following papers that have been discussed and cited in the manuscript. (1) Promising dawn in tumor microenvironment therapy: engineering oral bacteria. https://www.nature.com/articles/s41368-024-00282-3

(2) Association of vitamin K, fibre intake and progression of periodontal attachment loss in American adults. https://link.springer.com/article/10.1186/s12903-023-02929-9

(3) The role of extracellular vesicles in acquisition of resistance to therapy in glioblastomas. doi: 10.20517/cdr.2020.61

Manuscript briefly elaborated on specific research gaps or areas that require further investigation. This could help readers understand the potential directions for future studies. Please add future direction section or include it with conclusion.

The quality of Figures is poor. Please provide better quality images.

Conclusion: Avoid write in small paragraph. I think, you can arrange it one paragraph.

Please make sure to define each acronym at its first use. Check through the entire manuscript to make sure it is defined at the first use.

6. PLOS authors have the option to publish the peer review history of their article (what does this mean?). If published, this will include your full peer review and any attached files.

Reviewer #1: **Yes: **Dr. Surabhi Bajpai

Reviewer #2: No

---

## [Editor Report · Acceptance letter]

29 Aug 2024

PONE-D-24-24715 

PLOS ONE

Dear Dr. Melanitou, 

I'm pleased to inform you that your manuscript has been deemed suitable for publication in PLOS ONE. Congratulations! Your manuscript is now being handed over to our production team.

Kind regards, 

on behalf of

Dr. Nisha Singh 

Academic Editor

PLOS ONE